# Impaired Fear Extinction Recall in Serotonin Transporter Knockout Rats Is Transiently Alleviated during Adolescence

**DOI:** 10.3390/brainsci9050118

**Published:** 2019-05-22

**Authors:** Pieter Schipper, Paola Brivio, David de Leest, Leonie Madder, Beenish Asrar, Federica Rebuglio, Michel M. M. Verheij, Tamas Kozicz, Marco A. Riva, Francesca Calabrese, Marloes J. A. G. Henckens, Judith R. Homberg

**Affiliations:** 1Department of Cognitive Neuroscience, Donders Institute for Brain, Cognition and Behaviour, Radboud University Medical Center, Kapittelweg 29, 6525 EN Nijmegen, The Netherlands; ptrschipper@gmail.com (P.S.); deLeest@gmail.com (D.d.L.); Madder@hotmail.com (L.M.); beenish.phdns4@iiu.edu.pk (B.A.); frederica.rebuglio@unimi.it (F.R.); Michel.Verheij@radboudumc.nl (M.M.M.V.); Marloes.Henckens@radboudumc.nl (M.J.A.G.H.); 2Department of Pharmacological and Biomolecular Sciences, Universita’ degli Studi di Milano, 20133 Milan, Italy; paola.brivio@unimi.it (P.B.); m.riva@unimi.it (M.A.R.); francesca.calabrese@unimi.it (F.C.); 3Department of Clinical Genomics, Mayp Clinic, Rochester, MN 55905, USA; Tamas.Kozicz@radboudumc.nl

**Keywords:** serotonin transporter, rat, fear extinction, medial prefrontal cortex, NMDA, BDNF, adolescence, age

## Abstract

Adolescence is a developmental phase characterized by emotional turmoil and coincides with the emergence of affective disorders. Inherited serotonin transporter (5-HTT) downregulation in humans increases sensitivity to these disorders. To reveal whether and how *5-HTT* gene variance affects fear-driven behavior in adolescence, we tested wildtype and serotonin transporter knockout (5-HTT^−/−^) rats of preadolescent, adolescent, and adult age for cued fear extinction and extinction recall. To analyze neural circuit function, we quantified inhibitory synaptic contacts and, through RT-PCR, the expression of c-Fos, brain-derived neurotrophic factor (BDNF), and NDMA receptor subunits, in the medial prefrontal cortex (mPFC) and amygdala. Remarkably, the impaired recall of conditioned fear that characterizes preadolescent and adult 5-HTT^−/−^ rats was transiently normalized during adolescence. This did not relate to altered inhibitory neurotransmission, since mPFC inhibitory immunoreactivity was reduced in 5-HTT^−/−^ rats across all ages and unaffected in the amygdala. Rather, since mPFC (but not amygdala) c-Fos expression and NMDA receptor subunit 1 expression were reduced in 5-HTT^−/−^ rats during adolescence, and since PFC c-Fos correlated negatively with fear extinction recall, the temporary normalization of fear extinction during adolescence could relate to altered plasticity in the developing mPFC.

## 1. Introduction

Adolescence is a period of physical and brain maturation that is characterized by emotional turmoil and an increase in pervasive fears, and coincides with the emergence of anxiety and other affective disorders [1,2,3,4,5,6,7]. Recent data implicates organizational changes of the cognitive control circuitry regulating emotional behavior in this vulnerability during adolescence. More specifically, there is evidence for relative immaturity of the medial prefrontal cortex (mPFC) and its top-down control over subcortical areas mediating emotion and motivation such as the amygdala, whose development precedes that of the PFC [3]. According to the developmental mismatch hypothesis, the delayed maturation of the PFC in comparison to the amygdala results in a temporary imbalance between emotion and its regulatory processes [8]. However, there are substantial individual differences [9] in this transient “imbalance” during adolescence, and the underlying mechanisms and factors influencing the maturation process are not yet clear. As the PFC-amygdala circuit is dysfunctional in anxiety disorders [10] that frequently emerge during adolescence and often persist into adulthood [11], the understanding of the maturation of the PFC-amygdala circuit in healthy subjects is expected to inform the pathophysiology of stress-related neuropsychiatric disorders.

Appropriate PFC-amygdala circuit balance is critical for adequate extinction of fear. Previous research demonstrated that fear extinction is diminished in pre-adolescents and adolescents compared to adults, in both humans and animals [12,13,14]. This phenomenon only applies to fear that is cue-dependent and thus involving the PFC, whose activity—as assessed by c-Fos immunoreactivity—has been found to be reduced during adolescence compared to preadolescence and adulthood [12]. In contrast, improved extinction of contextual fear, as mediated by the hippocampus, is observed in adolescence compared to preadolescence and adulthood [15].

GABAergic inhibitory signaling plays an important role in this regulation of fear. The excitability of the basolateral amygdala (BLA), the amygdalar subnucleus responsible for maintaining the learned fear-association [16], is regulated by inhibitory signaling of local GABAergic interneurons [17], a mechanism by which fear and anxiety are attenuated [18]. Similarly, top-down control as mediated by the infralimbic cortex (IL) is modulated by GABAergic inhibition. The infralimbic cortex (IL) contributes to the inhibition of the fear response in the central amygdala (CeA) after successful fear extinction via its glutamatergic excitatory projections to the intercalated cells of the amygdala. IL function is regulated by excitatory inputs from several regions, including the BLA [19,20,21]. Although these projections are glutamatergic, their stimulation in vivo primarily inhibits neural activity in the PFC [22,23], which has been suggested to be caused by robust feedforward inhibition mediated by GABAergic interneurons [23,24]. Similar to the amygdala, some hippocampal projections may preferentially target IL interneurons, inhibiting IL output to downstream targets [25]. Besides this local inhibition, the IL also receives inhibitory innervations from the dorsal raphe nucleus (DRN) [26] as well as the basal forebrain [27]. Patients suffering from post-traumatic stress disorder, a disorder of aberrant fear extinction, are characterized by abnormalities in GABAergic signaling within the prefrontal cortex [28], implicating this local inhibitory circuit in its pathology. However, as of yet, the exact contribution of these inhibitory circuits to the impaired fear extinction in adolescents remains to be investigated.

Glutamate receptors represent another signaling system critical for the consolidation of extinction memories. Previous studies have demonstrated that the partial NMDA receptor agonist D-cycloserine (DCS) improves extinction retention in adolescent rats [13,29]. This implies that besides alterations in GABAergic signaling, a failure to recruit N-methyl-D-aspartate (NMDA) receptors may contribute to the impaired fear extinction during adolescence as well [12].

Brain-derived neurotrophic factor (BDNF) has also been implicated in fear circuitry maturation [30]. BDNF levels in the hippocampus peak during adolescence, suggesting that BDNF plays a key role in the maturation of subcortical regions. Furthermore, developmental studies utilizing a genetic *BDNF* single nucleotide polymorphism (Val66Met) knock-in mouse indicate that BDNF^Met/Met^ mice tested in preadolescence and early adolescence do not differ from wild-type controls regarding fear extinction, but show an impairment during adulthood. These results indicate that the impairment in cued fear extinction in BDNF^Met/Met^ mice emerges in a time frame corresponding to the transition from adolescence to adulthood and that BDNF may thus be critical in this developmental stage for appropriate circuit development.

Interestingly, the adolescent behavioral and supposed neural phenotype shows striking similarities to those seen in carriers of the low activity variant short (s) allele of the serotonin transporter linked polymorphic region (5-HTTLPR) in humans. Adult s-allele carriers, which presumably display increased extracellular serotonin levels, show increased acquisition [31] and reduced extinction [32] of conditioned fear, together with amygdala hyper-reactivity [33] and attenuated anatomical and functional coupling between the mPFC and amygdala [34,35]. Thus, the behavioral and brain phenotypes seen in adult carriers of the s-allele of the 5-HTTLPR may also imply a cortical-subcortical functional imbalance. Serotonin acts as a neurotrophic factor during development, and variations in serotonin availability occurring due to a limited availability of 5-HTT are thought to affect the development of circuits involved in the regulation of emotional behavior [36,37,38]. This poses the hypothesis that 5-HTTLPR may affect the development of the cortical-subcortical circuit, such that the transitions from preadolescence to adolescence, and from adolescence to adulthood are altered in 5-HTTLPR s-allele carriers.

Serotonin transporter knockout (5-HTT^−/−^) rats are used as a model organism for the 5-HTTLPR s-allele in humans and show many phenotypical similarities, both adaptive and maladaptive, to s-allele carriers [39]. Similar to humans and rodents during adolescence, as well as adult 5-HTTLPR s-allele carriers, 5-HTT^−/−^ rodents display impaired fear extinction (recall) [40,41,42,43,44,45,46]. Since 5-HTT^−/−^ rats display decreased inhibitory GABAergic control over excitatory neurons in the cortex during preadolescence [47], reduced expression of BDNF and GABA system components across development [48], altered NMDA receptor subunit expression in the PFC at adulthood [49], and an association with impaired fear extinction-reduced c-Fos expression in the IL [46], it is possible that the *5-HTT* genotype affects the development of the PFC–amygdala circuitry and thereby fear extinction recall across developmental stages.

Here, we employed a cued fear extinction paradigm to evaluate how differential 5-HTT expression affects the development of fear extinction learning and recall across adolescence using homozygous (5-HTT^−/−^) and heterozygous (5-HTT^+/−^) serotonin transporter knockout rats and compared them to wildtype animals (5-HTT^+/+^). We assessed the population of inhibitory cells in the IL and BLA by measuring the number of synaptic contacts expressing the inhibitory markers glutamic acid decarboxylase 65 and 67 (GAD65/67). Additionally, we assessed expression levels of BDNF, NMDA receptor subunits, and c-Fos in the PFC and amygdala at baseline and after fear extinction and fear extinction recall across ages in 5-HTT^+/+^ and 5-HTT^−/−^ rats.

## 2. Materials and Methods

### 2.1. Animals

All experiments were approved by the Committee for Animal Experiments of the Radboud University Nijmegen Medical Centre, Nijmegen, The Netherlands, and all efforts were made to minimize animal suffering and to reduce the number of animals used. Serotonin transporter knockout rats (Slc6a41Hubr) were generated on a Wistar background by N-ethyl-N-nitrosurea (ENU)-induced mutagenesis [50]. Experimental animals were derived from crossing heterozygous 5-HT transporter knockout (5-HTT^+/−^) rats that were outcrossed for at least 12 generations with wildtype Wistar rats obtained from Harlan Laboratories (Horst, The Netherlands). Ear punches were taken at the age of 21 days for genotyping, which was done by Kbiosciences (Hoddesdon, United Kingdom. Male adult 5-HTT^−/−^, 5-HTT^+/−^, and wildtype (5-HTT^+/+^) rats entered the experiment at p24 (preadolescent), p35 (adolescent), or p70 (adult). The adult animals were housed in pairs, while the adolescent and preadolescent animals were housed three per cage, in open cages. All animals had ad libitum access to food and water. A 12 h light–dark cycle was maintained, with lights on at 8:00 a.m. All behavioral experiments were performed between 8:00 a.m. and 6:00 p.m.

### 2.2. Apparatus

A 30.5 × 24.1 × 21 cm operant conditioning chamber (Model VFC-008, Med Associates) was used for fear conditioning and sham conditioning. The box was housed within a sound-attenuating cubicle and contained a white LED stimulus light, a white and near infrared house light, as well as a speaker capable of producing an 85 dB 2.8 kHz tone. The metal grid floor of the apparatus was connected to a scrambled shock generator (model ENV-412, Med Associates) configured to deliver shocks at 0.6 mA intensity. Fear extinction and extinction recall were tested in a novel context, in a novel room. The novel context consisted of a 25 × 25 × 30 cm Plexiglas cage, the bottom of which was covered with a +/− 0.5-cm-thick layer of black bedding. In this context, 85 dB (measured at the center of the floor) 2.8 kHz auditory stimuli were delivered through a set of external speakers.

### 2.3. Procedure

In total, 329 rats were exposed to behavioral testing. As genotypes of the animals at some ages were only known after completion of the protocol, relatively more 5-HTT^+/−^ animals were tested compared to 5-HTT^+/+^ and 5-HTT^−/−^ rats (*n*_5-HTT_^+/+^_-p24_ = 26, *n*_5-HTT_^+/+^_-p35_ = 30, *n*_5-HTT_^+/+^_-p70_ = 35, *n*_5-HTT_^+/−^_-p24_ = 51, *n*_5-HTT_^+/−^_-p35_ = 79, *n*_5-HTT_^+/−^_-p70_ = 32, *n*_5-HTT_^−/−^_-p24_ = 25, *n*_5-HTT_^−/−^_-p35_ = 21, *n*_5-HTT_^−/−^_-p70_ = 30). On the day on which the animals entered the experiment (p24 for the preadolescent group, p35 for the adolescent group, and p70 for the adult group), the animals were habituated to the conditioning context for 10 minutes. Twenty-four hours after habituation, animals were given a cued fear conditioning session. Fear conditioning began with a 2-minute habituation period, followed by 5 instances of a 30-second 85 dB 2.8 kHz auditory stimulus co-terminating with a 1-second 0.6 mA foot shock, followed by a 1-minute inter-trial interval. Twenty-four, 48, and 72 hours after conditioning, fear extinction and two sessions of extinction recall were given, respectively. Thus, extinction learning and extinction recall (2×) were assessed on three consecutive days. In each of these sessions, rats were exposed to a 2-minute habituation period, after which 24 20-second presentations of the auditory stimulus were given, with an inter-trial interval of 5 seconds. Sessions were recorded, and freezing was automatically assessed by a software program (see below). For the conditioning and the habituation to the fear conditioning chamber, the apparatus was cleaned before and after each animal using a tissue slightly dampened with 70% EtOH. Water was used for cleaning in between the extinction and extinction recall sessions.

### 2.4. Assessment of Behavior

Time spent freezing during the conditioning session was not assessed, as previous work as indicated no differences between genotypes in the acquisition of fear memory [46]. For assessing the time spent freezing during extinction learning and both extinction recall sessions, we used the Ethovision 9.0 behavioral software package (Noldus Information Technology B.V., Wageningen, the Netherlands). Freezing was determined using the Activity Monitor feature of the software package. The threshold for pixel change between frames was set between 0.05 and 0.09% (depending on the specific camera in use, but not different between groups). Automatic assessment was compared to manually scored samples in in total 696 samples of 20 seconds, derived from 29 extinction sessions by two different observers blind to the genotype of the animal, and proved to be a reliable assessment of freezing behavior (correlation between manual and automatic outcomes: *r* = 0.7397). To analyze fear extinction learning, extinction sessions were divided into 6 blocks representing the average freezing responses to 4 auditory cue presentations each. Average freezing to all auditory cue presentations during the recall sessions was used as index for fear extinction recall.

Since 5-HTT^+/−^ and 5-HTT^+/+^ showed a comparable behavioral profile, we focused on 5-HTT^−/−^ and 5-HTT^+/+^ rats during subsequent histological and molecular studies aiming to understand the mechanisms underlying the genotype × age effects.

### 2.5. GAD65/67 Immunostaining

The immunostaining procedure was adopted from Olivier et al. (2008) and Nonkes et al. (2010) [51,52]. Ninety minutes following either the extinction learning session or the second extinction recall session, a subset of the rats (*n* = 5, randomly selected) were anesthetized and perfused transcardially with 0.1 mol/L PBS, pH 7.3, followed by 4% paraformaldehyde dissolved in 0.1 mol/L phosphate buffer (PB), pH 7.2. The pressure of the perfusion was reduced for the preadolescent rats. Perfusion continued until signs of successful perfusion were observed (shaking limbs, stiff cheeks, etc.). Subsequently, the brains were removed from the skull and post-fixed overnight in 4% paraformaldehyde at 4 °C. Before sectioning, the brains were cryoprotected with 30% sucrose in 0.1 mol/L PB. Forty-micrometer-thick brain sections were cut on a freezing microtome and collected in six parallel series in 0.1 mol/L PBS containing 0.1% sodium azide. One series from each rat was used for every staining. The free-floating sections were washed three times in PBS and preincubated with 0.3% perhydrol (30% H2O2, Merck, Darmstadt, Germany) for 30 min. After washing three times in PBS, the sections were presoaked for 30 min in an incubation medium consisting of PBS with 0.1% bovine serum albumin and 0.5% Triton X-100. The sections were then incubated with goat anti-GAD65/67, 1:2000 (Santa Cruz Biotechnology Inc., Santa Cruz, CA, USA), overnight on a shaker, at room temperature, and consecutively incubated for 90 min at room temperature with biotinylated donkey-anti-goat (Jackson Immuno Research Laboratories, West Grove, PA, USA) diluted 1:1500 in incubation medium and for 90 min at room temperature with ABC-elite, diluted 1:800 in PB (Vector Laboratories, Burlingame, CA, USA). Between incubations, sections were rinsed three times with PBS. The GAD65/67–antibody peroxidase complex was made visible using 3,3-diaminobenzidine tetrahydrochloride staining. Sections were incubated for 10 min in a chromogen solution consisting of 0.02% 3,3-diaminobenzidine tetrahydrochloride and 0.03% nickel–ammonium sulfate in 0.05 mol/L Tris-buffer (pH 7.6) and subsequently for 10 min in chromogen solution containing 0.006% hydrogen peroxide. This resulted in a blue–black staining. The sections were then rinsed three times in PBS and mounted on gelatin chrome alum-coated glass slides, dried overnight in a stove at 37 °C, dehydrated in an increased series of ethanol, cleared in xylene, embedded with Entellan (Merck), and coverslipped.

### 2.6. Quantification

Numbers of GAD65/67-immunopositive granules, representing inhibitory synaptic contacts, were quantified using the software program Fiji ImageJ, a public domain image-processing program (http://rsb.info.nih.gov/ij/) [53]. Granules were counted in the IL in equally framed sections across groups at 2.20 from Bregma at ×40 magnification using an Axio Imager.A2 microscope (Zeiss, Oberkochen, Germany). BLA GAD65/67 immunoreactivity was measured in sections at −1.88 mm from Bregma at ×40 magnification. The results for each subject are expressed as the total amount of immunopositive granules counted in a standardized sample area measuring 281.6 × 211.2 um within each section.

### 2.7. Gene Expression Analyses

The remaining animals were sacrificed by rapid decapitation at 90 minutes following either the extinction learning session or the second extinction recall session. Brains were rapidly removed from the skull and quick-frozen on dry ice and stored at −80 ℃ until further processing. 

Brains from WT and 5-HTT^−/−^ rats were sectioned into 220 µm coronal slices on a Leica CM3050 S Research Cryostat (Leica Biosystems, Amsterdam, the Netherlands), with a chamber temperature of −12 °C and an object temperature of −10 °C, after which regions of interest were punched out. To be able to relate gene expression profiles following extinction (recall) to basal gene expression patterns, additional naïve control WT and 5-HTT^-/-^ brains were obtained and processed in a similar fashion (*n*_5-HTT_^+/+^_-p24_ = 6, *n*_5-HTT_^+/+^_-p35_ = 7, *n*_5-HTT_^+/+^_-p70_ = 7, *n*_5-HTT_^−/−^_-p24_ = 7, *n*_5-HTT_^−/−^_-p35_ = 7, *n*_5-HTT_^−/−^_-p70_ = 6). Medial prefrontal cortex punches were taken bilaterally with a 1.0 mm diameter hollow needle from 8 subsequent slices (Bregma ≈ 3.70:2.20 mm), for a total of 32 punches (prelimbic and infralimbic cortex were punched bilaterally and punches combined to obtain sufficient amounts of material for gene expression analyses). Likewise, 8–10 1.0 mm diameter punches were taken from the bilateral amygdala (Bregma ≈ −2.30: −3.30 mm).

Total RNA was isolated by a single step of guanidinium isothiocyanate/phenol extraction using PureZol RNA isolation reagent (Bio-Rad Laboratories, Italy) according to the manufacturer’s instructions and quantified by spectrophotometric analysis. Following total RNA extraction, the samples were processed for real-time polymerase chain reaction (RT-PCR) to assess total BDNF, NR1, NR2A, and c-Fos mRNA expression. An aliquot of each sample was treated with DNase to avoid DNA contamination. RNA was analyzed by TaqMan qRT-PCR instrument (CFX384 real time system, Bio-Rad Laboratories, Segrate, Italy) using the iScriptTM one-step RT-PCR kit for probes (Bio-Rad Laboratories, Segrate, Italy). Samples were run in 384 well formats in triplicate as multiplexed reactions with a normalizing internal control (β-actin). Primers sequences (Table 1) used were purchased from Eurofins MWG-Operon.

Thermal cycling was initiated with an incubation at 50 °C for 10 min (RNA retrotranscription) and then at 95 °C for 5 min (TaqMan polymerase activation). After this initial step, 39 cycles of PCR were performed. Each PCR cycle consisted of heating the samples at 95 °C for 10 s to enable the melting process and then for 30 s at 60 °C for the annealing and extension reactions. A comparative cycle threshold method was used to calculate the relative target gene expression [54] 

### 2.8. Statistics

All statistical analyses were performed using SPSS Statistics version 24.0 (SPSS Inc., IBM, Armonk, NY, USA). Data are presented as mean ± standard error of the mean (SEM). Behavioral data were analyzed using a repeated measures analysis of variance (ANOVA), whereas the immunohistochemical and gene expression were analyzed using a 2-way ANOVA, with genotype and age (preadolescent, adolescent, adult) as between-subject factors. Statistical testing on the latter was performed on obtained deltaCT values, whereas data are plotted as fold-change expression levels relative to the preadolescent 5-HTT^+/+^ group. For Pearson correlation analyses between freezing and neural measures, we averaged freezing rates observed during all cue presentations to a single measure. Probability *p*-values of less than 0.05 were considered significant. Bonferroni correction was applied to correct for multiple testing in post hoc tests.

## 3. Results

### 3.1. Freezing Behavior

*Baseline freezing.* To measure baseline freezing, we assessed freezing during the 2-minute stimulus free period preceding the first extinction session. Freezing in response to the novel context was significantly affected by age (F_(2,319)_ = 41.016, *p* < 0.001), but not genotype (F_(2,319)_ = 1.745, *p* = 0.176), and no significant genotype × age interaction was found (F_(4,319)_ < 1) (Figure 1). Bonferroni post hoc analysis revealed that adolescent animals froze more upon novel context exposure than adult animals (*p* < 0.001), while preadolescent animals froze more than adolescent and adult animals (both *p* < 0.001).

*Fear extinction learning.* In the extinction learning session, freezing during the cue presentations reduced over blocks (F_(5,324)_ = 145.945, *p* < 0.001), and this reduction (i.e. the speed of extinction learning) was dependent on both age (block × age interaction; F_(10,650)_ = 3.607, *p* < 0.001) and genotype (block × genotype interaction; F_(20,650)_ = 3.458, *p* < 0.001), but not on a genotype × age interaction (F_(20,1308)_ < 1) (Figure 1). Exploration of the genotype effect through post hoc tests revealed that 5-HTT^−/−^ rats showed slower extinction learning than both 5-HTT^+/−^ and 5-HTT^+/+^ rats (both *p* < 0.001), whereas 5-HTT^+/−^ and 5-HTT^+/+^ animals showed similar extinction rates (*p* = 0.653). Exploration of the age effect revealed significant differences in extinction learning curves between all three ages, which seemed to be driven by slower extinction in pre-adolescent compared to adolescent rats (*p* = 0.006) and lower initial freezing (in block 1) of adult rats compared to preadolescent rats (*p* = 0.008). There were no age effects within the genotypes (*p* > 0.1).

*First fear extinction recall.* Total freezing during the first extinction recall session was used as a behavioral indicator of the recall of the extinction memory acquired during the first fear extinction learning session. We observed a main effect of genotype (F_(2,144)_ = 4.051, *p* = 0.019), a trend-level significant main effect of age (F_(2,144)_ = 2.910, *p* = 0.058) and a genotype × age interaction for this parameter (F_(4,144)_ = 2.747, *p* = 0.031) (Figure 1). The latter appeared to be driven by a significant effect of genotype in the preadolescent (F_(2,46)_ = 6.016, *p* = 0.005), but not the adolescent (F_(2,52)_ = 1.401, *p* = 0.255) and adult animals (F_(2,46)_ = 2.254, *p* = 0.116). The genotype effect in the preadolescent group was driven by 5-HTT^−/−^ rats, which froze significantly more than 5-HTT^+/−^ (*p* = 0.012) and 5-HTT^+/+^ (*p* = 0.007) animals, while freezing was not different between 5-HTT^+/−^ and 5-HTT^+/+^ animals (*p* = 1.000). When comparing age effects in genotype groups we observed that fear extinction recall was significantly affected by age in 5-HTT^−/−^ rats (F_(2,35)_ = 60.527, *p* = 0.004), but not 5-HTT^+/+^ (F_(2,35)_ < 1) and 5-HTT^+/−^ (F_(2,74)_ < 1 rats. The age effect in 5-HTT^−/−^ rats was attributed to improved recall during adolescence compared to preadolescence (*p* = 0.004) and adulthood (*p* = 0.049), in the absence of a difference between the latter two groups (*p* = 0.471).

*Second fear extinction recall*. We found a main effect of genotype (F_(2,142)_ = 8.601, *p* < 0.001), age (F_(2,142)_ = 10.756, *p* < 0.001), and genotype × age interaction (F_(4,142)_ = 2.921, *p* = 0.023) in freezing behavior during the second extinction recall session (Figure 1). Here, we found a significant effect of genotype in the preadolescent (F_(2,44)_ = 7.334, *p* = 0.002) and the adult group (F_(2,46)_ = 6.115, *p* = 0.004), but again not in the adolescent animals (F_(2,52)_ < 1). 5-HTT^−/−^ rats froze more than 5-HTT^+/−^ and 5-HTT^+/+^ animals in both the preadolescent (*p* = 0.001 and *p* = 0.016, respectively) and the adult (*p* = 0.005 and *p* = 0.057 respectively) age groups, while freezing between 5-HTT^+/−^ and wildtype animals was not different in either age group (both *p*-values = 1.000). When comparing age effects in genotype groups, we observed that fear extinction recall was significantly affected by age in 5-HTT^−/−^ rats (F_(2,35)_ = 75.819, *p* = 0.002), but not 5-HTT^+/+^ rats (F_(2,35)_ = 1.286, *p* = 0.289). In 5-HTT^−/−^ rats, reduced freezing was observed during adolescence as compared to preadolescence (*p* = 0.002), but not adulthood (*p* = 0.125), whereas freezing at these latter two ages did not differ significantly (*p* = 0.125). In 5-HTT^+/−^ rats, a significant effect of age was found (F_(2,72)_ = 20.583, *p* = 0.037), caused by improved fear extinction with age (resulting in a significant difference in freezing during recall in preadolescence vs. adulthood (*p* = 0.036), whereas the other comparisons were non-significant (all *p*-values > 0.27)). As all significant genotype effects were driven by aberrant behavior of the 5-HTT^−/−^ rats, further neural analyses focused on the comparison of these genotypes with their 5-HTT^+/+^ counterparts.

In Appendix A, the freezing per genotype across the three ages is depicted, and Appendix A depicts the freezing across blocks during the recall sessions.

### 3.2. GAD65/67 Immunoreactivity 

*Infralimbic cortex.* The number of GAD65/67 immunopositive granules in the IL was significantly affected by genotype (F_(1,24)_ = 14.326, *p* = 0.001), but not age (F_(2,24)_ = 2.110, *p* = 0.143), and no genotype × age interaction could be detected (F_(2,24)_ = 1.222, *p* = 0.312, Figure 2). The number of granules expressing GAD65/67 was significantly reduced in 5-HTT^−/−^ animals compared to 5-HTT^+/+^ animals (*p* = 0.001). Although the effect of genotype did not significantly differ between age groups, post hoc testing revealed the most prominent effects of genotype in preadolescent rats (*p* < 0.001), whereas adolescent and adult rats did not display significant differences between genotypes (*p*-values > 0.3).

*Basolateral amygdala.* No effects of genotype (F_(1,24)_ < 1) or age (F_(1,24)_ < 1), nor a genotype × age interaction (F_(2,24)_ = 1.583, *p* = 0.226), were found in the number of GAD65/67 immuno-positive granules in the BLA (Figure 2).

### 3.3. Gene Expression Levels Neuronal Plasticity and Activity Genes

*Basal expression. mPFC.* In the mPFC of naive control animals (Figure 3, upper panel), c-Fos expression was affected by genotype (F_(1,32)_ = 16.321, *p* < 0.001) and age (F_(2,32)_ = 3.502, *p* = 0.042), but not by a genotype × age interaction (F_(2,32)_ = 1.828, *p* = 0.177). These effects appeared to be driven by significantly lower c-Fos expression levels in 5-HTT^−/−^ compared to 5-HTT^+/+^ rats (*p* < 0.001), whereas adolescent animals tended to display increased expression compared to pre-adolescent (*p* = 0.036), but not adult (*p* = 0.236) rats. BDNF expression was also dependent on genotype (F_(1,33)_ = 29.072, *p* < 0.001) and age (F_(2,33)_ = 27.108, *p* < 0.001), without displaying a genotype × age interaction (F_(2,33)_ < 1). Additionally, BDNF levels were significantly lower in 5-HTT^−/−^ compared to 5-HTT^+/+^ rats (*p* < 0.001), whereas adolescent rats displayed the highest expression (both *p*-values < 0.001), whereas adult rats displayed higher levels than preadolescent rats (*p* = 0.007). NR1 levels only depended on the age of the rat (F_(2,33)_ = 71.644, *p* < 0.001), with again the adolescent rats displaying the highest expression (both *p*-values < 0.001), and adult rats displaying higher levels than preadolescent rats (*p* = 0.020). NR2A levels were characterized by a main effect of age (F_(2,32)_ = 113.835, *p* < 0.001) and a genotype × age interaction (F_(2,32)_ = 8.020, *p* = 0.002). Similarly to NR1 and BDNF, NR2A expression levels were highest in adolescence (both *p*-values < 0.001), and adult rats showed higher NR2A expression than pre-adolescent rats (*p* = 0.001). Moreover, in adolescence, 5-HTT^−/−^ rats displayed significantly higher NR2A expression levels compared to 5-HTT^+/+^ rats (*p* < 0.001), whereas no differences between genotypes were observed at preadolescence (*p* = 0.165) and adulthood (*p* = 0.666).

*Amygdala.* In the amygdala (Figure 3, lower panel), c-Fos expression was modulated by age (F_(2,34)_ = 7.090, *p* = 0.003), but not genotype (F_(1,34)_ < 1) nor a genotype × age interaction (F_(2,34)_ = 1.171, *p* = 0.322). This age effect was driven by a significantly higher expression in adolescent compared to preadolescent (*p* = 0.005) and adult rats (*p* = 0.011), whereas no differences between these latter age groups were found (*p* = 1.000). BDNF expression in the amygdala was modulated by a genotype × age interaction (F_(2,32)_ = 6.067, *p* = 0.006), but no main effects (both *p*-values > 0.2). Further testing suggested that this interaction was driven by lower amygdala BDNF expression in pre-adolescent and adolescent 5-HTT^−/−^ rats compared to WTs (*p* = 0.041 and *p* = 0.046 respectively), whereas adult 5-HTT^−/−^ rats tended to display increased amygdala BDNF expression (*p* = 0.069). Amygdala NR1 expression was not modulated by genotype, age (both F-values < 1), or a genotype × age interaction (F_(2,32)_ = 1.059, *p* = 0.359), whereas NR2A expression was different for the distinct age groups (F_(2,32)_ = 11.156, *p* < 0.001), without a significant effect of genotype (F_(1,32)_ < 1) or genotype × age interaction (F_(2,32)_ = 2.371. *p* = 0.110). Further testing revealed that pre-adolescent rats displayed lower amygdala NR2A expression compared to adolescent and adult rats (both *p*-values = 0.001), whereas the latter two age groups were not different (*p* = 1.000).

### 3.4. Gene Expression following Fear Extinction Learning

*mPFC.* Levels of c-Fos expression in the mPFC following extinction learning (Figure 4, upper panel) were dependent on the rats’ age (F_(2,36)_ = 6.182, *p* = 0.005), but not on genotype (F_(1,36)_ = 1.032, *p* = 0.317) or on the genotype × age interaction (F_(2,36)_ < 1). Preadolescent rats showed lower c-Fos expression than adolescent (*p* = 0.014) and adult (*p* = 0.002) animals, whereas adolescent and adult animals displayed similar levels (*p* = 0.850). mPFC BDNF expression following extinction was also dependent on age (F_(2,39)_ = 7.507, *p* = 0.002) and showed a trend towards an effect of genotype (F_(1,39)_ = 3.107, *p* = 0.086), without displaying a genotype × age interaction (F_(2,39)_ = 1.185, *p* = 0.316). Similar to naïve animals, BDNF levels were highest in adolescent rats (*p* < 0.001 and *p* = 0.003 compared to preadolescent and adult rats, respectively), whereas no differences were observed between adult and preadolescent rats (*p* = 0.502). Adolescent 5-HTT^−/−^ rats showed lower mPFC BDNF expression than 5-HTT^+/+^ rats (*p* = 0.014), while no significant differences were observed at the other ages (both *p*-values > 0.5). NR1 levels only depended on the age of the rat (F_(2,40)_ = 30.131, *p* < 0.001), with again the adolescent rats displaying the highest expression (both *p*-values < 0.001), and levels in preadolescent and adult rats not differing (*p* = 0.206). Similarly, mPFC NR2A expression following extinction was characterized by a main effect of age (F_(2,39)_ = 36.840, *p* < 0.001), but no effect of genotype or genotype × age interaction (both F-values < 1). Again, expression levels were highest in adolescence (both *p*-values < 0.001), and adult rats showed higher NR2A expression than pre-adolescent rats (*p* = 0.031). No correlations were observed between basal or cue-induced freezing and mPFC expression levels.

*Amygdala.* In the amygdala (Figure 4, lower panel), c-Fos expression following extinction learning was modulated by age (F_(2,40)_ = 5.918, *p* = 0.006) and a genotype × age interaction (F_(2,40)_ = 3.870, *p* = 0.029), whereas the main effect of genotype did not reach significance (F_(1,40)_ = 3.092, *p* = 0.086). This age effect was driven by a higher expression in the adolescent compared to preadolescent amygdala (*p* = 0.017), whereas neither age group significantly differed from adults (*p* = 0.547 and *p* = 0.379 respectively). The interaction was driven by a significant genotype effect in preadolescent rats, with WTs showing lower expression (*p* = 0.047), whereas no differences were observed at the other ages (both *p*-values > 0.22). Amygdala BDNF expression was also dependent on age (F_(2,37)_ = 5.158, *p* = 0.011), without the effect of genotype nor the interaction (both F-values < 1). BDNF levels were higher in the adult compared to the preadolescent (*p* = 0.053) and adolescent (*p* = 0.006) amygdala, whereas the latter were not different from each other (*p* = 1.000). Similarly, amygdala NR1 expression following extinction depended on age (F_(2,40)_ = 4.992, *p* = 0.012), but not genotype or a genotype × age interaction (both F-values < 1), with adult rats displaying the same expression levels as the preadolescent rats (*p* = 0.149) but higher levels compared to adolescent rats (*p* = 0.007). These groups did not differ from each other (*p* = 0.927)). Amygdala NR2A expression was not affected by age, genotype, or their interaction (all F-values < 1).

Correlational analyses across all ages and genotypes related both amygdala BDNF and NR1 levels to basal anxiety, with lower expression levels following testing being related to higher freezing during the habituation period (BDNF: *r*(43) = 0.307, *p* = 0.045; NR1: *r*(46) = 0.310, *p* = 0.036) (Appendix A). Moreover, amygdala BDNF was negatively related to cue-induced freezing during the extinction session (*r*(43) = 0.440, *p* = 0.003) (Appendix A).

### 3.5. Gene Expression following Fear Extinction Recall

*mPFC.* Following the last fear extinction recall session, expression levels of c-Fos in the mPFC (Figure 5, upper panel) were modulated by the rats’ age (F_(2,25)_ = 4.993, *p* = 0.015), genotype (F_(1,25)_ = 13.612, *p* = 0.001), and a genotype × age interaction (F_(2,25)_ = 4.046, *p* = 0.030). Further testing revealed that WT rats showed highest expression levels in adolescence (*p* < 0.001 and *p* = 0.001 compared to preadolescent and adult animals, respectively) and higher levels in adult compared to preadolescent rats (*p* = 0.043). No such effect of age was observed in 5-HTT^−/−^ rats (all *p*-values = 1.000), resulting in significantly higher mPFC c-Fos expression in WT compared to 5-HTT^−/−^ rats during adolescence (*p* = 0.013), but not preadolescence (*p* = 0.778) or adulthood (*p* = 0.075). mPFC BDNF expression following extinction recall was modulated by a genotype × age interaction (F_(2,28)_ = 5.397, *p* = 0.010), without the main effects of age (F < 1) or genotype (F_(1,28)_ = 1.084, *p* = 0.307). This interaction was caused by a significant effect of age in 5-HTT^+/+^ rats (F_(2,13)_ = 6.174, *p* = 0.013) that was absent in 5-HTT^−/−^ rats (F_(2,15)_ = 1.672, *p* = 0.221), resulting in a significant effect of genotype only at adult age (*p* = 0.005, other *p*-values > 0.2), with 5-HTT^+/+^ rats displaying lower BDNF expression. Additionally, mPFC NR1 levels following extinction recall were modulated in a genotype × age manner (F_(2,28)_ = 4.034, *p* = 0.029), without main effects of age or genotype (both F-values < 1). Preadolescents (*p* = 0.960) of both genotypes showed similar NR1 expression. Adolescent 5-HTT^−/−^ rats were characterized by lower NR1 expression compared to their 5-HTT^+/+^ counterparts (*p* = 0.048), whereas adult 5-HTT^−/−^ rats tended to show higher expression (*p* = 0.081). NR2A expression levels were characterized by a genotype × age interaction as well (F_(2,28)_ = 4.080, *p* = 0.028), without significant effects of age (F < 1) or genotype (F_(1,28)_ = 1.920, *p* = 0.177). Post hoc testing only revealed a significant effect of genotype during adulthood, when mPFC NR2A expression in response to extinction recall was significantly increased in 5-HTT^−/−^ compared to 5-HTT^+/+^ rats (*p* = 0.034). No significant differences were found during preadolescence and adolescence between 5-HTT^−/−^ compared to WT rats (*p* = 0.106 and *p* = 0.137, respectively). 

Correlational analyses across all ages and genotypes revealed that mPFC c-Fos expression was significantly related to the amount of cue-induced freezing during this last extinction recall session (*r*(31) = 0.366, *p* = 0.043), with reduced c-Fos levels relating to increased freezing, reflecting impaired extinction recall (Appendix A).

*Amygdala.* In the amygdala (Figure 5, lower panel), c-Fos expression following extinction recall was not modulated by age, genotype, or a genotype × age interaction (all F-values < 1). Amygdala BDNF expression revealed a trend towards an age × genotype interaction (F_(2,28)_ = 3.242, *p* = 0.054), without a main effect of genotype (F_(1,28)_ = 1.111, *p* = 0.301) or age (F_(2,28)_ = 2.450, *p* = 0.105). Exploratory post hoc tests revealed a significant reduction in amygdala BDNF expression during preadolescence in 5-HTT^−/−^ compared to 5-HTT^+/+^ rats (*p* = 0.017) that was not observed at other ages (*p*-values > 0.6). Amygdala NR1 expression following extinction recall only revealed a significant effect of age (F_(2,28)_ = 5.178, *p* = 0.012), but not of genotype (F_(1,28)_ = 1.106, *p* = 0.302) or of their interaction (F_(2,28)_ = 1.403, *p* = 0.262), with adult rats displaying higher expression levels compared to preadolescent (*p* = 0.035) and adolescent rats (*p* = 0.015), whereas these latter groups did not differ from each other (*p* = 1.000). Amygdala NR2A expression only revealed trends for a reduced expression in 5-HTT^−/−^ rats across ages (F_(1,28)_ = 3.871, *p* = 0.056) and an increase with age (F_(2,28)_ = 2.662, *p* = 0.087), without interaction (F < 1). No significant correlations between amygdala gene expression levels and freezing during extinction recall were observed.

## 4. Discussion

Here, we confirm that fear extinction recall is impaired in 5-HTT^−/−^ rats, an established and often replicated phenomenon [44,45,55,56], as is extinction learning in rats of this genotype [46]. Strikingly, an effect of age on fear extinction recall was seen only in 5-HTT^−/−^ rats, which enjoyed a transient normalization (i.e. improvement) of fear extinction recall during adolescence. Whereas augmented fear extinction learning seems to be responsible for the improved fear extinction recall observed in 5-HTT^−/−^ rats during adolescence, age × genotype effects on learning rates failed to reach significance. The number of GAD65/67 positive synaptic contacts, indicative of inhibitory regulation, was decreased in the IL of 5-HTT^−/−^ rats, regardless of age, and no clear effect of age or genotype were seen on the number of GAD65/67 positive synaptic contacts in the BLA. In naïve rats, we observed increases in BDNF, NR1, and NR2A expression levels in the mPFC, and in c-Fos in the mPFC and amygdala, during adolescence. Furthermore, BDNF levels were reduced in 5-HTT^−/−^ rats across all ages. While no genotype × age interactions were observed following fear extinction learning, fear extinction recall was associated with a genotype × age interaction for NR1, NR2A, and c-Fos in the mPFC. These data suggest that specifically (glutamatergic) plasticity changes in the mPFC contribute to the temporary normalization of fear extinction recall in 5-HTT^−/−^ rats during adolescence.

A number of developmental abnormalities arising from 5-HTT abolishment have been described in the literature. The development of several motor and sensory functions, namely reflexes, motor coordination and olfactory discrimination, is delayed in 5-HTT^−/−^ rats but normalized upon reaching adulthood [57]. Remarkably, other deficiencies seen in adult 5-HTT^−/−^ animals, i.e. impaired object recognition, object directed behavior, and sensorimotor gating, do not arise until after adolescence [57]. The present results suggest that the abnormal emotional profile seen in 5-HTT^−/−^ rats is subject to a nonlinear developmental trajectory as well, implying that 5-HTT abolishment influences neural maturation depending on the developmental phase and locus. The finding of the transiently alleviated recall of fear extinction during adolescence in 5-HTT^−/−^ rats suggests that the pacing of development of cortical and subcortical regions may be altered in these rats. Congruent with our findings, a study in 5-HTT^−/−^ mice has demonstrated that increased anxiety, another hallmark trait of the 5-HTT^−/−^ rodent phenotype, is not present during adolescence [58].

This study does not replicate findings from other studies that suggest fear extinction recall deficits in adolescent animals and humans with normal 5-HTT expression [12,13], as our results indicate that, in 5-HTT^+/+^ animals, fear extinction recall is not significantly affected by age. We corroborate findings of another study, in which extinction learning was found to be similar between adolescent and adult C57BL/6J mice [59]. Differences in the details of the experimental procedures may crucially determine whether an effect of age presents itself. For instance, the experiments may differ in the degree to which contextual cues from the conditioning session are present during the extinction, which determines the additional involvement of the hippocampus on fear expression and extinction [60]. This variability in the reported findings necessitates additional investigation towards the exact circumstances under which adolescent fear extinction (recall) is impaired.

The inhibitory immunoreactivity in the IL as assessed by immunohistochemistry is reduced in 5-HTT^−/−^ rats across all age groups. This finding is in line with previous observations of reduced inhibitory synapses onto cortical excitatory neurons in preadolescent 5-HTT^−/−^ rats [47]. Previous work has, however, associated *increased* inhibitory synaptic transmission onto IL projection neurons with impaired retrieval of extinction memory by inhibiting the consolidation of extinction [61], which contrasts our observation of reduced inhibition in the genotype group with poorest extinction recall. Yet, here we did not determine the class of neurons targeted by these inhibitory contacts, leaving the possibility of reduced inhibition of local interneurons in 5-HTT^−/−^ rats open. In any case, the observed reduction in inhibitory synapses appears to remain stable across the development from preadolescence to adulthood, making it unlikely that altered development of prefrontal inhibition contributing to the remarkable development of fear extinction behavior seen in these animals.

Under basal conditions, in naive rats, c-Fos, BDNF, NR1, and NR2A gene expression levels in the mPFC were highest during adolescence, indicating that adolescence is indeed a critical period of mPFC development. For NR2A, we additionally observed that levels were highest during adolescence in 5-HTT^−/−^ rats. The peak in NMDA receptor expression may relate to pruning (removal of synapses), known to occur during adolescence and to be NMDA-receptor-dependent [62]. Increased c-Fos expression levels in the PFC during adolescence may reflect a compensatory attempt of the mPFC to retain control over the amygdala, while the lower c-Fos expression levels in 5-HTT^−/−^ rats across ages may correspond to the reduced prefrontal cortical top-down control over the amygdala as reported for human 5-HTTLPR s-allele carriers [34]. The peak in BDNF levels during adolescence is in line with previous observations [30]. We also replicated previous observations of reduced BDNF expression in the PFC of 5-HTT^−/−^ regardless of age [48,63,64]. In the amygdala, c-Fos levels were found to peak during adolescence, which potentially reflects the increased activity of this area due to reduced prefrontal top-down control [4]. However, c-Fos remained high during adulthood, which might reflect the completion of amygdala maturation during adolescence. Amygdala BDNF levels were reduced in 5-HTT^−/−^ rats during preadolescence and adolescence, in line with the overall decreased BDNF levels in these rats found previously [48,63,64], but BDNF levels tended to be increased in 5-HTT^−/−^ rats during adulthood. For NR1, no genotype and age effects were observed, and for NR2A there was a decrease in expression in preadolescent rats. These data show that the mPFC and amygdala mature at different paces and through different plasticity routes.

BDNF, NR1, and NR2A expression levels in the mPFC after fear extinction learning largely recapitulated the baseline findings in naive rats, suggesting that extinction learning does not change the expression of these plasticity factors. During the recall test, however, we observed that (over all animals and ages combined) c-Fos expression in the mPFC was negatively correlated with cue-induced freezing. This implies that impaired extinction recall is associated with reduced prefrontal cortex activity and thereby cognitive control over the emotional response. This finding is in line with the study of Patwell et al. [12] reporting a link between impaired extinction recall and reduced c-Fos expression in the IL in adolescent animals. Nonetheless, the increased mPFC c-Fos expression in 5-HTT^+/+^ adolescents is quite remarkable. It is important to note that we combined IL and PrL tissue for gene expression analyses, raising the possibility that the increase in c-Fos expression in adolescent 5-HTT^+/+^ rats is due to increased c-Fos expression in the PrL. The function of this is open to speculation. As this observation does not result in lower freezing levels in adolescent 5-HTT^−/−^ rats, other neuroplasticity changes in the mPFC or amygdala might counteract this effect. BDNF and NR2A were increased in adult 5-HTT^−/−^ rats specifically, which thereby seem to be unrelated to the temporary improvement in fear extinction in adolescent 5-HTT^−/−^ rats. We furthermore observed that adolescent 5-HTT^−/−^ rats display lower levels of NR1 in the mPFC. The essential NR1 subunit of the NMDA receptor expressed in excitatory prefrontal cortical neurons has been shown to decrease fear generalization [65]. If NMDAR-dependent neural signaling in the mPFC is a component of a neural mechanism for disambiguating the meaning of fear signals, our finding may point towards a temporary improvement in the interpretation of the fear-predicting cue during adolescence in 5-HTT^−/−^ rats, allowing the animals to discriminate the fear and safety better. We did not explicitly assess fear generalization in this study, but the measure that comes closest is baseline freezing observed prior to the tone presentations in the extinction recall sessions. Interestingly, baseline freezing prior to extinction recall was modulated by genotype, with 5-HTT^−/−^ rats displaying higher freezing than the other groups (Appendix A). Thus, 5-HTT^−/−^ rats showed higher baseline freezing levels as well as reduced mPFC NR1 expression in adulthood. However, these measures did not significantly correlate (*p* > 0.15), leaving our interpretation still speculative. In the amygdala, c-Fos expression tended to peak in adolescence independently of genotype, which thereby follows the pattern observed in the mPFC. None of the other genes assessed displayed an expression pattern that followed the age- and genotype-dependent changes in freezing during extinction learning. This implies that the amygdala does not play a key role in the temporary disappearance of genotype effects on freezing during extinction learning in adolescence. We did observe that BDNF and NR1 expression significantly correlated with baseline freezing behavior. Specifically, lower expression of both NR1 and BDNF in the amygdala was associated with more freezing during the habituation period. Furthermore, amygdala BDNF was negatively related to cue-induced freezing during the extinction session. Amygdala BDNF has been demonstrated to facilitate fear learning [66], which appears incongruent with our observation. Potentially, lower BDNF levels in the amygdala mediated unconditioned fear in this study. Overall, our data suggest that the temporary normalization of fear extinction recall in 5-HTT^−/−^ rats during adolescence relates to neuroplasticity changes in the mPFC, whereas the amygdala seems to exert more generalized (genotype-independent) effects on the freezing response.

Some limitations of the study require attention. The quantified granules in the IL are hypothesized to represent synaptic contacts. However, without performing a functional tracer study, it is not possible to determine the source of the GABAergic inputs. In addition, it is not certain that these synaptic terminals interface with neurons that are functional within the circuitry driving fear expression and extinction. In addition, animals that had undergone one and three days of fear extinction were pooled to determine GAD65/67 immunoreactivity in the IL and BLA to obtain sufficient statistical power for a comparison. Since GAD65/67 expression is influenced by recent fear conditioning, it is possible that levels of expression were affected by this variation in time between conditioning and sacrifice of the animal. However, all GAD65/67 positive synaptic contacts were included in the assessment regardless of expression level; given the high signal to the background ratio of the DAB-Ni, variations in expression due to the varying regimes of fear conditioning is unlikely to have affected the findings. Furthermore, because CT values were too different between the obtained from the naïve, extinction learning, and extinction recall group, we did not express the gene expression changes after extinction as the percentage of baseline gene expression in the naïve animals. For RT-PCR, we punched the whole mPFC, while we studied the IL part of the mPFC in the immunohistochemical study. This was necessary to obtain a sufficient amount of tissue for the PCRs and to reduce gene expression variance due to variations in the precise positioning of the punch needle. As a consequence, it is possible that differential gene expression in the IL and PrL diluted the effects we observed for the whole mPFC. Another limitation is that we did not measure freezing during conditioning during acquisition. We previously observed no genotype differences during fear conditioning in adults [46]. However, we do not know whether genotype differences are also absent during preadolescence and adulthood. Since freezing during Blocks 1–4 was not different between genotypes during the fear memory recall/extinction session, it is not likely there were genotype differences in freezing during conditioning. As yet another limitation, visual observation of Figure 1 implies that the increased freezing during Session 2 and 3 is due to an extinction learning deficit during Session 1. However, the observed age × genotype effects as observed during the extinction recall sessions seemed to result from altered recall of extinction. Nonetheless, additional differences in extinction learning between genotypes and ages cannot be ruled out. Finally, housing conditions varied between the age groups; although no animals were kept in isolation, preadolescent and adolescent animals were housed with more cage mates than adults for practical and ethical reasons. Although this aspect is often overlooked in animal research concerning stress and psychiatric illness, social elements in housing conditions have been shown to influence emotional behavior [67] and are known to be especially influential and instrumental to psychiatric wellbeing during adolescence [68].

## 5. Conclusions

In conclusion, the present findings show that the influence of genetic reduction of *5-HTT* expression on the development of fear extinction recall manifests in a non-linear pattern, temporarily normalizing during adolescence, to become deficient again at adulthood. This discovery raises as many questions as it answers; delayed or aberrant maturation of cortical or subcortical regions or interconnecting tracts is a likely cause but exploiting this finding for therapeutic benefit will require further specification of their nature and functional implications. The anatomical and functional development of excitatory neurons in the IL projecting to the amygdala are of particular interest for future study. An in vivo electrophysiology or calcium imaging study in which single neurons or populations of neurons are followed across the different stages from fear conditioning to extinction and extinction recall would be enlightening. As it stands, the data suggest that reduced inhibitory signaling within the IL and temporary altered excitatory signaling in the mPFC represent potential causes for the impaired control over the amygdala seen in individuals with reduced expression of 5-HTT and its temporary normalization during adolescence.

## Figures and Tables

**Figure 1 brainsci-09-00118-f001:**
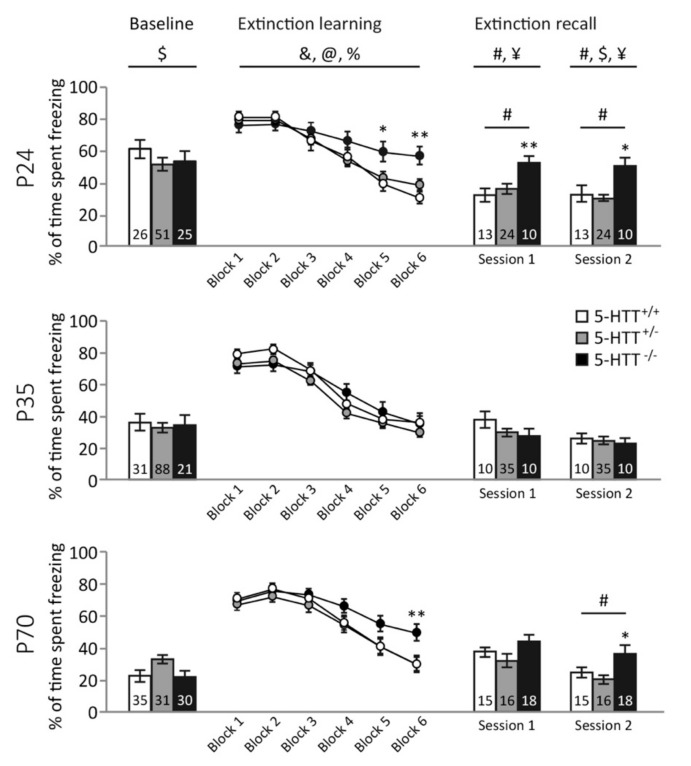
Fear conditioning behavioral data across extinction learning and the two extinction recall sessions. Freezing during the 2-minute stimulus free baseline period preceding extinction learning decreased across age in all genotypes. Fear extinction learning is impaired in preadolescent 5-HTT^−/−^ rats, normalized in this genotype during adolescence, and impaired again in adulthood. Fear extinction recall is impaired in preadolescent 5-HTT^−/−^ rats, normalized in this genotype during adolescence, and impaired again in adulthood. Data are expressed as the mean % of time spent freezing during stimulus presentations ± standard error of the mean. #: a significant effect of genotype (*p* < 0.05); $: a significant effect of age (*p* < 0.05); ¥: a significant age × genotype interaction (*p* < 0.05); *: a significant post hoc difference between 5-HTT^−/−^ vs. 5-HTT^+/−^ and/or 5-HTT^+/+^ rats (*p* < 0.05); **: a significant post hoc difference between 5-HTT^−/−^ vs. 5-HTT^+/−^ and/or 5-HTT^+/+^ rats (*p* < 0.05); & a significant effect of extinction block (*p* < 0.05); @ a significant age × block interaction (*p* < 0.05); % a significant genotype × block interaction (*p* < 0.05).

**Figure 2 brainsci-09-00118-f002:**
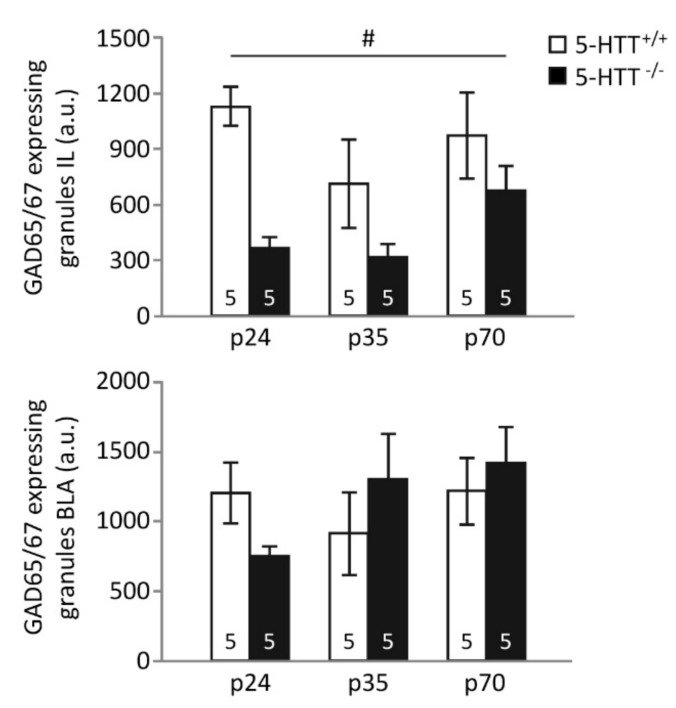
GAD65/67 immunoreactivity in the infralimbic cortex (IL) and basolateral amygdala (BLA) of preadolescent (p24), adolescent (p35), and adult (p70) 5-HTT^−/−^ and 5-HTT^+/+^ rats. GAD 65/67 immunoreactivity is significantly reduced in preadolescent, adolescent, and adult 5-HTT^−/−^ animals in the IL, but not BLA. #: a significant effect of genotype (*p* < 0.05).

**Figure 3 brainsci-09-00118-f003:**
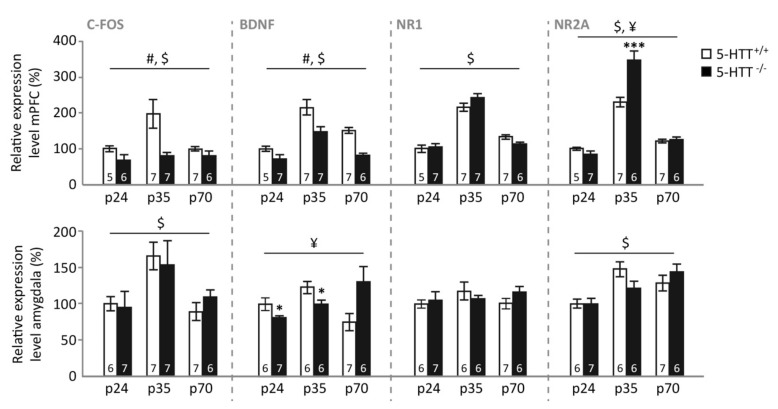
Relative expression levels of c-Fos, BDNF, NR1, and NR2A in the medial prefrontal cortex (mPFC) and amygdala of naive preadolescent (p24), adolescent (p35), and adult (p70) 5-HTT^−/−^ and 5-HTT^+/+^ rats. #: a significant effect of genotype (*p* < 0.05); $: a significant effect of age (*p* < 0.05); ¥: a significant age × genotype interaction (*p* < 0.05); *: a significant post hoc difference between 5-HTT^−/−^ vs. age-matched 5-HTT^+/+^ rats (* *p* < 0.05; *** *p* < 0.001).

**Figure 4 brainsci-09-00118-f004:**
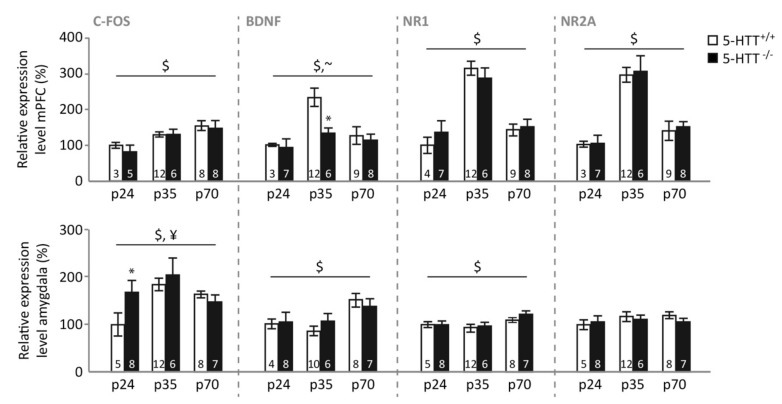
Relative expression levels of c-Fos, BDNF, NR1, and NR2A in the medial prefrontal cortex (mPFC) and amygdala of 5-HTT^−/−^ and 5-HTT^+/+^ rats following fear extinction learning during preadolescence (p24), adolescence (p35), and adulthood (p70). #: a significant effect of genotype (*p* < 0.05); ~: a trend-level significant effect of genotype (*p* = 0.086); $: a significant effect of age (*p* < 0.05); ¥: a significant age × genotype interaction (*p* < 0.05); *: a significant post hoc difference between 5-HTT^−/−^ vs. age-matched 5-HTT^+/+^ rats (*p* < 0.05).

**Figure 5 brainsci-09-00118-f005:**
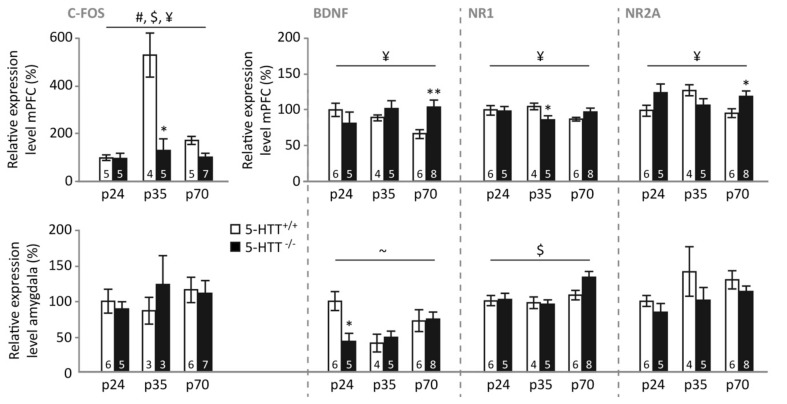
Relative expression levels of c-Fos, BDNF, NR1, and NR2A in the medial prefrontal cortex (mPFC) and amygdala of 5-HTT^−/−^ and 5-HTT^+/+^ rats following the second session of fear extinction recall during preadolescence (p24), adolescence (p35), and adulthood (p70). #: a significant effect of genotype (*p* < 0.05); $: a significant effect of age (*p* < 0.05); ¥: a significant age × genotype interaction (*p* < 0.05); ~: a trend-level significant age × genotype interaction (*p* = 0.054); *: a significant post hoc difference between 5-HTT^−/−^ vs. age-matched 5-HTT^+/+^ rats (* *p* < 0.05; ** *p* < 0.01).

**Table 1 brainsci-09-00118-t001:** Sequences of forward and reverse primers and probes used in real-time PCR analyses and purchased from Eurofins MWG-Operon.

Gene	Forward Primer	Reverse Primer	Probe
***BDNF tot***	AAGTCTGCATTACATTCCTCGA	GTTTTCTGAAAGAGGGACAGTTTAT	TGTGGTTTGTTGCCGTTGCCAAG
***NR1***	TCATCTCTAGCCAGGTCTACG	CAGAGTAGATGGACATTCGGG	TGGGAGTGAAGTGGTCGTTGGG
***NR2A***	GCACCAGTACATGACCAGATTC	ACCAGTTTACAGCCTTCATCC	CGTCCAACTTCCCGGTTTTCAAGC
***c-Fos***	TCCTTACGGACTCCCCAC	CTCCGTTTCTCTTCCTCTTCAG	TGCTCTACTTTGCCCCTTCTGCC
***β*** ***-actin***	CACTTTCTACAATGAGCTGCG	CTGGATGGCTACGTACATGG	TCTGGGTCATCTTTTCACGGTTGGC

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
