# Peer review of "Impaired Fear Extinction Recall in Serotonin Transporter Knockout Rats Is Transiently Alleviated during Adolescence"

_brainsci, 2019, doi:10.3390/brainsci9050118_

Round 1
Reviewer 1 Report
This manuscript is well written and is on a topic of interest in the field of fear regulation across development. The authors have identified a gap in the literature and have nicely written the introduction. The rationale was clear and I was left thinking it was remarkable that no-one had yet studied how serotonin transporter knockout affects fear extinction during development and neural activity in the prefrontal cortex and amygdala. There is a considerable amount of work that has gone into this study with three ages, three genotypes, three time points of neural measures, two brain regions, and many neural measures. The authors have definitely gone above and beyond here searching for neural explanations for their behavioural results! Their results in wildtype naive animals are nice additions to the literature on neural changes in the BLA and PFC across development. The most difficult thing in reporting this study was perhaps conveying and integrating the many results in a meaningful way, particularly as some findings were unexpected. Line 532 was a nice summary. The authors have done a good job at discussing the findings of the study in terms of past literature which is well-cited (although see point 9 below) but I felt that some claims about their results were overstated.
As an example of where the interpretation isn’t clear, the statement that adolescent 5-HTT-/- rats had lower NR1 in the mPFC (after extinction retention?) and this reflected decreased generalized seemed a bold claim as the effect, although significant, seemed small. Further, no genotype differences were found in the BLA in adults despite the mean difference appearing larger (but with more variability?). How do the non-linear NR1 results in naive animals or at extinction relate to the linear decrease in fear generalization/baseline anxiety?
One interesting behavioural finding was that there was a clear decrease in fear generalization to the novel context at baseline with increasing age. The juvenile animals showed very high levels of baseline freezing. This is an interesting result in this study. It helps differentiates the results from the current study to other previous work on extinction across development. There is some discussion that this reflects anxiety. I wondered if perhaps the contexts for conditioning and extinction were more similar than used in previous studies and therefore the younger animals are generalizing across the contexts substantially more than in other studies examining cued fear extinction. Would this contribute to the differences in extinction retention across development and especially the lack of extinction retention deficit in adolescent animals reported here? The authors cited work that adolescents show enhanced extinction of context fear during adolescence. It may be the case that there is more extinction of context-US associations occurring in this study (despite the “novel” context) than in previous studies where there is low baseline freezing in the novel context even in juveniles. In helping address this question more details of how the contexts differed could be provided in the methods section (e.g., is the bottom of the novel context only bedding - no grid floors in this chamber?). Were they any differences in the lighting or visual cues in this chamber to differentiate it from the conditioning context? This issue briefly comes up in the discussion at Line 499 but there is no comment about the current study relative to others.
The authors stated that Bonferroni correction was applied to correct for multiple testing in post hoc tests. Yet there are several post hoc comparisons that do not appear to have been Bonferroni corrected. For example, line 295 p = .036, line 351, line 431 p=.048, line 450.
I would encourage the authors to show a figure of all correlations that are discussed. For instance, I would like to see a scatter plot of mPFC cFos being inversely correlated to cue-induced freezing during the last extinction session. The cFos findings in IL are particularly striking in Fig 5a. It is interesting that these findings are completely different to those of Pattwell and colleagues [12] who I think examined cFos after extinction recall.
How was freezing which was presented across blocks correlated with neural measures? Was it averaged across the session? This should be clearer.
It is excellent that there are measures of various markers at several time points. However, a downside of presenting the naive, extinction, and recall gene expression data separately is that we cannot get a sense of whether cFos (or any other measure) is increased or decreased at various stages of the experiment. With the cFos data I wondered whether extinction training was upregulating cFos activity in all ages relative to naive so that the high baseline activity in the wildtype adolescents is abolished or instead whether all levels have returned to baseline. I wondered whether it is the former but of course this is speculative. There is some electrophysiological data that EPSC amplitude is non-specifically increased in the medial prefrontal cortex of adolescents compared to other ages [12]. Would this increase in basal synaptic transmission help interpret the high levels of neuronal activity indicated by cFos in this region in naive animals? Similarly, it would be so informative to know whether the cFos data for mPFC at extinction recall is a return to “baseline” levels or not (i.e., some upregulation of neuronal activity at another extinction session). I’m not necessarily suggesting more experimental work for this study but this could be considered for future work.
The limitations section mention GAD analyses were pooled across days of extinction for power. The sample sizes were 5 per group. Yet in the naive animal analyses of gene expression the sample sizes were only slightly larger of 6 or 7 but presumably sufficiently powered?
In terms of minor comments:
1. In line 53 the authors may consider including which groups they are comparing adolescents to.
2. In line 121 do they authors mean that ear punches were taken at weaning which occurred at 21 days of age or were they taken 21 days after weaning (which occurred at an unspecified age). As written it appears like the latter but the authors could mean the former.
3. Line 156&157 indicate that conditioning session wasn’t assessed as previous work indicated no genotype differences in fear acquisition. This seemed to oppose a statement in line 84 that adults have increased acquisition. Are there differences in humans [21] versus findings in rats [36]?
4. How many (or proportion of) samples were cross-scored by a trained observer?
5. Line 175 “a part of the rats” – do you mean a subset of rats
6. Were juveniles perfused with 400ml paraformaldehyde even though their body weight was smaller than adults?
7. How many brain sections from the IL and BLA were counted per region? What was the area of the framed sections for IL images?
8. Line 397 – p value is .0149 – was this deemed significant? What was the alpha value?
9. Ref 49 line 497: I would encourage the authors to revise this statement. Both early adolescent (4 week old) and peri-adolescent (6 week old) mice showed impaired extinction relative to adults (8 weeks old).
Author Response
Dear editor,
Thank you for handling our manuscript entitled ”Impaired fear extinction recall in serotonin transporter knockout rats is transiently alleviated during adolescence”. We appreciate the constructive and elaborative comments of the reviewers, which helped us to improve the manuscript. Please find our detailed responses to the reviewer comments below. Besides the changes in response to reviewers we noted that the immunostainings for GAD67 did not involve cell bodies, but granules/puncta, hence synaptic contacts. We revised the manuscript accordingly. Changes in the manuscript are marked in red. We carefully proof-read our manuscript. We hope that our manuscript is now suitable for publication in Brain Sciences.
Rev. 1
This manuscript is well written and is on a topic of interest in the field of fear regulation across development. The authors have identified a gap in the literature and have nicely written the introduction. The rationale was clear and I was left thinking it was remarkable that no-one had yet studied how serotonin transporter knockout affects fear extinction during development and neural activity in the prefrontal cortex and amygdala. There is a considerable amount of work that has gone into this study with three ages, three genotypes, three time points of neural measures, two brain regions, and many neural measures. The authors have definitely gone above and beyond here searching for neural explanations for their behavioural results! Their results in wildtype naive animals are nice additions to the literature on neural changes in the BLA and PFC across development. The most difficult thing in reporting this study was perhaps conveying and integrating the many results in a meaningful way, particularly as some findings were unexpected. Line 532 was a nice summary. The authors have done a good job at discussing the findings of the study in terms of past literature which is well-cited (although see point 9 below), but I felt that some claims about their results were overstated.
As an example of where the interpretation isn’t clear, the statement that adolescent 5-HTT-/- rats had lower NR1 in the mPFC (after extinction retention?) and this reflected decreased generalized fear seemed a bold claim as the effect, although significant, seemed small. Further, no genotype differences were found in the BLA in adults despite the mean difference appearing larger (but with more variability?). How do the non-linear NR1 results in naive animals or at extinction relate to the linear decrease in fear generalization/baseline anxiety?
Reply: Thank you for the kind words. As the reviewer mentions, we obtained a lot of data and sometimes struggled to explain all findings in the light of past literature. Apologies if we overstated our interpretations. Regarding NR1 expression in the mPFC during the last extinction recall test (Figure 5), we found a significant reduction in 5-HTT-/-rats during adolescence (p = 0.048). We did not find any significant genotype effect (F(1, 28)= 1.106, p = 0.302) or genotype x age interaction (F(2, 28)= 1.403, p = 0.262) in NR1 expression in the BLA during the last extinction recall test. Therefore, no further testing for genotype effects at the separate age groups was warranted. Our interpretation of decreased mPFC NR1 expression contributing to fear generalization was based on previous literature indicating a role for NR1 in excitatory prefrontal cortical neurons in reducing fear generalization (Vieiraet al. 2015). We did not explicitly assess fear generalization in this study, but the measure that comes closest is baseline freezing observed prior to the tone presentations in the extinction recall sessions. Therefore, we tested whether baseline freezing during extinction recall session 2 was significantly affected by age and genotype. Similarly to baseline freezing levels observed prior to fear extinction, these data revealed a strong effect of age (F(2,142)= 9.063, p < 0.001), with preadolescent animals showing highest freezing levels (p < 0.001 compared to adolescents; p = 0.002 compared to adults), whereas adolescent animals did not differ from adults (p = 1.000). However, interestingly, baseline freezing prior to extinction recall was also modulated by genotype (F(2,142)= 3.970, p = 0.021), with 5HTT-/-rats displaying higher freezing than the other groups (F(1,107)= 8.281, p = 0.005, compared to 5HTT+/-; F(1,107)= 2.983, p = 0.089, compared to 5HTT+/+), whereas 5HTT+/-and 5HTT+/+did not significantly differ from each other in freezing behaviour (F < 1). Thus, 5HTT-/-rats showed higher baseline freezing levels as well as reduced mPFC NR1 expression in adulthood. However, these measures did not significantly correlate (p > 0.15), leaving our interpretation still speculative. We included these findings in the manuscript (Figure S3) and rephrased our claims accordingly.
One interesting behavioural finding was that there was a clear decrease in fear generalization to the novel context at baseline with increasing age. The juvenile animals showed very high levels of baseline freezing. This is an interesting result in this study. It helps differentiate the results from the current study to other previous work on extinction across development. There is some discussion that this reflects anxiety. I wondered if perhaps the contexts for conditioning and extinction were more similar than used in previous studies and therefore the younger animals are generalizing across the contexts substantially more than in other studies examining cued fear extinction. Would this contribute to the differences in extinction retention across development and especially the lack of extinction retention deficit in adolescent animals reported here? The authors cited work that adolescents show enhanced extinction of context fear during adolescence. It may be the case that there is more extinction of context-US associations occurring in this study (despite the “novel” context) than in previous studies where there is low baseline freezing in the novel context even in juveniles. In helping address this question more details of how the contexts differed could be provided in the methods section (e.g., is the bottom of the novel context only bedding - no grid floors in this chamber?). Were they any differences in the lighting or visual cues in this chamber to differentiate it from the conditioning context? This issue briefly comes up in the discussion at Line 499 but there is no comment about the current study relative to others.
Reply: The contexts used for extinction learning and recall were actually very different, both in terms of cage (its size, floor, walls, and light conditions) and room. The extinction context existed of an empty, white-colored wall Plexiglas cage, with sawdust on the bottom. Hence, it is not likely that the high baseline freezing in the juveniles reflects contextual freezing, because the extinction context had never been associated with shocks or conditioned stimuli predicting shock during the baseline measurement prior to the extinction training. Rather, it may be that juveniles respond differently to a completely novel environment than older animals. Since ‘anxiety’ might not be the proper term to use, we rephrased the baseline freezing results to “baseline freezing”. The information on the contexts is elaborated upon in the revised section “Apparatus” in the material and methods section.
The authors stated that Bonferroni correction was applied to correct for multiple testing in post hoc tests. Yet there are several post hoc comparisons that do not appear to have been Bonferroni corrected. For example, line 295 p = .036, line 351, line 431 p=.048, line 450.
Reply: All p-values reported in the manuscript are the Bonferroni-corrected p-values (if applicable). This we explicitly mention in materials and methods section.
I would encourage the authors to show a figure of all correlations that are discussed. For instance, I would like to see a scatter plot of mPFC cFos being inversely correlated to cue-induced freezing during the last extinction session. The cFos findings in IL are particularly striking in Fig 5a. It is interesting that these findings are completely different to those of Pattwell and colleagues [12] who I think examined cFos after extinction recall.
Reply: We included the scatter plots of all reported significant correlations as a supplemental figure to the manuscript (Figure S3). Note that we opted for the depiction of relative expression levels (compared to preadolescent 5HTT+/+levels) instead of deltaCT values on which we actually performed the statistical testing, as this makes the graphs more intuitive. Whereas we do not replicate the findings of Patwell et al. [12] in terms of overall impaired fear extinction during adolescence, we do replicate the same association as Patwell and colleagues report of impaired extinction recall being linked to reduced cFos expression in the IL. Patwell at al. observed reduced IL cFos expression following extinction recall in adolescent animals, which were characterized by impaired extinction learning and recall. We report on a significant negative correlation between mPFC cFos expression and extinction recall (cue-induced freezing) as well. However, we do agree with the reviewer that the observation of increased mPFC cFos expression in WT adolescents is remarkable, which we now comment on in the discussion section of the revised manuscript.
How was freezing which was presented across blocks correlated with neural measures? Was it averaged across the session? This should be clearer.
Reply: To increase comprehension of the data we indeed averaged freezing rates observed during all cue-presentations to a single measure. We now explicitly mention this in the materials and methods section of the revised manuscript.
It is excellent that there are measures of various markers at several time points. However, a downside of presenting the naive, extinction, and recall gene expression data separately is that we cannot get a sense of whether cFos (or any other measure) is increased or decreased at various stages of the experiment. With the cFos data I wondered whether extinction training was upregulating cFos activity in all ages relative to naive so that the high baseline activity in the wildtype adolescents is abolished or instead whether all levels have returned to baseline. I wondered whether it is the former but of course this is speculative. There is some electrophysiological data that EPSC amplitude is non-specifically increased in the medial prefrontal cortex of adolescents compared to other ages [12]. Would this increase in basal synaptic transmission help interpret the high levels of neuronal activity indicated by cFos in this region in naive animals? Similarly, it would be so informative to know whether the cFos data for mPFC at extinction recall is a return to “baseline” levels or not (i.e., some upregulation of neuronal activity at another extinction session). I’m not necessarily suggesting more experimental work for this study but this could be considered for future work.
Reply: We agree with the reviewer that this point is a limitation of the study: we could not compare the extinction (recall) data to the data from the naïve animals, because the raw CT values were too different from each other. This we also mentioned in the discussion section of the manuscript. Thus, we can only look at relative differences between genotype within the same experimental condition and further interpretations would be pure speculation. c-Fos expression could be modulated in both glutamatergic and GABAergic neurons, we cannot say. We agree that the work is not finished yet, and have provided a suggestion for future research in the conclusion: “An in vivo electrophysiology or calcium imaging study in which single neurons or populations of neurons are followed across the different stages from fear conditioning to extinction and extinction recall would be a great thing to do as future study”.
The limitations section mention GAD analyses were pooled across days of extinction for power. The sample sizes were 5 per group. Yet in the naive animal analyses of gene expression the sample sizes were only slightly larger of 6 or 7 but presumably sufficiently powered?
Reply: Since we were able to replicate literature findings on age-dependent differences in the expression of BDNF (see line 530) and see clear differences in gene expression during adolescence compared to preadolescence and adult, we believe the power was sufficient.
In terms of minor comments:
In line 53 the authors may consider including which groups they are comparing adolescents to.
Reply: Thank you, we have added this information.
In line 121 do they authors mean that ear punches were taken at weaning which occurred at 21 days of age or were they taken 21 days after weaning (which occurred at an unspecified age). As written it appears like the latter but the authors could mean the former.
Reply: The ear punches were taken at the age of 21 days. We made this now clearer.
Line 156&157 indicate that conditioning session wasn’t assessed as previous work indicated no genotype differences in fear acquisition. This seemed to oppose a statement in line 84 that adults have increased acquisition. Are there differences in humans [21] versus findings in rats [36]?
Reply: This is a good point, as this indeed would suggest differences between rats and humans. One study (Johnson et al., 2019; not by us) has reported on enhanced fear conditioning in 5-HTT-/-rats, but these effects were already present during the first tone, i.e. before the rats received any shock. We do not know how to interpret this, because this contrasts with what we found before; no genotype differences in fear conditioning (Shan et al., 2018). There are some differences in the protocol used, such as a shorter inter-trial interval in our study. We are not certain whether we increased freezing during conditioning in 5-HTT-/-rats is a correct finding, although it would be in line with the human findings.
How many (or proportion of) samples were cross-scored by a trained observer?
Reply: We measured the correlation between automatic and manual scoring of freezing in in total 696 samples of 20 seconds, derived from 29 extinction sessions, by two different observers. This information is now included in the manuscript.
Line 175 “a part of the rats” – do you mean a subset of rats.
Reply: Yes, that is correct. We added n=5, to make it more concrete.
Were juveniles perfused with 400ml paraformaldehyde even though their body weight was smaller than adults?
Reply: We perfused the preadolescent rats with paraformaldehyde at a much-reduced pressure. We continued with the perfusion until signs of successful perfusion were observed (shaking limbs, stiff jaw, etc.), instead of using a fixed amount of paraformaldehyde per animal. Perfusion was followed by overnight post-fixation of the brain. This we clarified in the materials and methods section.
How many brain sections from the IL and BLA were counted per region? What was the area of the framed sections for IL images?
Reply: We counted one section for the IL and one section for the BLA, using a sample rectangle that was placed at the same location for each IL and BLA section. This is mentioned in the materials and methods section. The size of the rectangle was 281.6 x 211.2 um within. This is now also mentioned in the results section.
Line 397 – p value is .149 – was this deemed significant? What was the alpha value?
Reply: The reviewer is right in stating that a p-value of 0.149 is not significant. We rephrased the text accordingly.
Ref 49 line 497: I would encourage the authors to revise this statement. Both early adolescent (4 week old) and peri-adolescent (6 week old) mice showed impaired extinction relative to adults (8 weeks old).
Reply: Thank you, we revised this statement.
Reviewer 2 Report
Schipper et al. test fear extinction in serotonin transporter knockout (5-HTT-/-), wild-type (5-HTT+/+), and heterozygous (5-HTT+/-) rats of different ages (preadolescent, adolescent, adult). The authors also assay a range of molecular changes (BDNF, GAD, C-fos, NR1 and NR2) in the medial prefrontal cortex and amygdala of naïve and trained rats at each age. The developmental effects of serotonin on extinction are understudied and the topic is of interest to a wide audience. Unfortunately, important statistical analyses are missing and the data do not support many of the claims made by the authors. Specific concerns are outlined below:
1. On lines 466-467,tThe authors state that they “confirm that fear extinction recall is impaired in 5-HT+/+ rats.” However, this conclusion cannot be drawn from the data shown. There is a within-session extinction impairment in KO rats at p24, as demonstrated by higher levels of freezing in blocks 5 and 6. Given that the KO rats freeze significantly more than other two groups at the end of the first day of extinction, it is expected that they would freeze more the next day, because they are still learning to extinguish. This higher level of freezing the next day may have nothing to do with impaired extinction recall. To test extinction recall, rats should have been tested until they reached control levels of freezing on the previous day of extinction.
2. Line graphs showing freezing responses across blocks of tone trials should be shown for each day of extinction recall. There may be effects of genotype during early tone exposure on these days that are relevant.
3. Freezing behavior during the first 2min pre-CS period on the first day of extinction training was used to measure baseline anxiety. This is not an accurate way to measure anxiety, as any freezing responses likely reflect generalization of fear from the training to the testing context. This is a learned response and not an anxiety response. A more appropriate way to measure baseline anxiety would have involved testing open field or elevated plus maze behavior prior to fear conditioning.
4. Freezing to the testing context (baseline freezing) at p24 is exceptionally high in all genotypes. Levels of freezing do not appear to change during the first block of tone presentations. We therefore cannot know if rats are freezing to the tone or the context. Similarly, are animals extinguishing to the tone or context? This is a big limitation of this study.
5. The authors should show all correlational analyses in a figure.
6. The authors should specify the sex of the animals that were tested.
7. The authors make several claims about the effects of age on extinction. To visualize these effects, the authors should graph the behavior of wild-type rats at each age, so direct comparisons can be made. An ANOVA on that data set should be done to confirm an effect of age.
8. On lines 472-473, the authors state that “the number of GAD65/67 positive cells…was decreased in the IL of 5-HTT-/- rats, regardless of age…” It is clear from Figure 2, that there is no significant difference between 5-HT+/+ and 5-HTT-/- rats in IL GAD at p70. The main effect of genotype is being driven by data from rats in the p24 and p35 groups. Similar mistakes are made throughout the manuscript.
9. The inter-trial interval (ITI) during extinction testing was exceptionally short (5 seconds). A longer ITI is typical in rat studies and would have improved within-session extinction.
10. The authors should have assessed freezing during acquisition, as this can affect later levels of fear expression and extinction. Although the authors refer to previous work indicating that genotype does not affect learning, it is unclear whether that previous study tested the same three age groups.
11. It is difficult to follow the effects of the molecular analyses. The results of all post-hoc tests (multiple comparisons) should be clearly indicated with asterisks in the graphs.
Author Response
Dear editor,
Thank you for handling our manuscript entitled ”Impaired fear extinction recall in serotonin transporter knockout rats is transiently alleviated during adolescence”. We appreciate the constructive and elaborative comments of the reviewers, which helped us to improve the manuscript. Please find our detailed responses to the reviewer comments below. Besides the changes in response to reviewers we noted that the immunostainings for GAD67 did not involve cell bodies, but granules/puncta, hence synaptic contacts. We revised the manuscript accordingly. Changes in the manuscript are marked in red. We carefully proof-read our manuscript. We hope that our manuscript is now suitable for publication in Brain Sciences.
Rev. 2
Schipper et al. test fear extinction in serotonin transporter knockout (5-HTT-/-), wild-type (5-HTT+/+), and heterozygous (5-HTT+/-) rats of different ages (preadolescent, adolescent, adult). The authors also assay a range of molecular changes (BDNF, GAD, C-fos, NR1 and NR2) in the medial prefrontal cortex and amygdala of naïve and trained rats at each age. The developmental effects of serotonin on extinction are understudied and the topic is of interest to a wide audience. Unfortunately, important statistical analyses are missing and the data do not support many of the claims made by the authors. Specific concerns are outlined below:
On lines 466-467, The authors state that they “confirm that fear extinction recall is impaired in 5-HT+/+ rats.” However, this conclusion cannot be drawn from the data shown. There is a within-session extinction impairment in KO rats at p24, as demonstrated by higher levels of freezing in blocks 5 and 6. Given that the KO rats freeze significantly more than the other two groups at the end of the first day of extinction, it is expected that they would freeze more the next day, because they are still learning to extinguish. This higher level of freezing the next day may have nothing to do with impaired extinction recall. To test extinction recall, rats should have been tested until they reached control levels of freezing on the previous day of extinction.
Reply: We agree with the reviewer that the on average increased freezing levels during the first fear extinction recall session could potentially be explained by impaired initial fear extinction learning. However, if this would be the case, one might expect the 5-HTT-/-rats to catch up in fear extinction during the presentation of subsequent cues, as tested for in the second and third fear extinction session. However, in these two sessions, no significant block x genotype effects were observed (recall session 1: p = 0.123; recall session 2; p = 0.549), nor block x age x genotype interactions (recall session 1: p = 0.272; recall session 2; p = 0.847), indicating that genotype did not modulate extinction learning across blocks in these sessions. However, there were significant main effects of genotype (recall session 1: p = 0.018; recall session 2: p = 0.001), as well as genotype x age effects (recall session 1: p = 0.032; recall session 2: p = 0.039), suggesting that genotype affected freezing behaviour across the whole session, independent of block. Therefore, we believe that the increased freezing behavior as observed in pre-adolescent and adult 5-HTT-/-rats is not the mere result of impaired extinction learning, but instead indicates additional effects of impaired extinction recall. To support this point, we included an additional figure (Figure S2) to the revised manuscript depicting the freezing across blocks in the recall sessions.
Line graphs showing freezing responses across blocks of tone trials should be shown for each day of extinction recall. There may be effects of genotype during early tone exposure on these days that are relevant.
Reply: As mentioned in our previous answer, we have included these data as a new supplemental figure to the revised manuscript (Figure S2).
Freezing behavior during the first 2min pre-CS period on the first day of extinction training was used to measure baseline anxiety. This is not an accurate way to measure anxiety, as any freezing responses likely reflect generalization of fear from the training to the testing context. This is a learned response and not an anxiety response. A more appropriate way to measure baseline anxiety would have involved testing open field or elevated plus maze behavior prior to fear conditioning.
Reply: The fear extinction context used was completely different from the fear conditioning context. The extinction context existed of an empty, white-colored wall, Plexiglas cage, with sawdust on the bottom, which was placed in a different room than the FC box (we have added this information to the revised manuscript). Hence, the 2 min pre-CS freezing rather involves a response to novelty, and it is not likely that this behavior would be very different from behavioral responses observed in an open field or elevated plus maze. The animals did not experience shock or conditioned cue exposure in this new context before. Therefore, we deem it unlikely that contextual freezing is involved. Nonetheless, we decided to remove the term ‘anxiety’ and refer to it as ‘baseline freezing’; exactly what it is.
Freezing to the testing context (baseline freezing) at p24 is exceptionally high in all genotypes. Levels of freezing do not appear to change during the first block of tone presentations. We therefore cannot know if rats are freezing to the tone or the context. Similarly, are animals extinguishing to the tone or context? This is a big limitation of this study.
Reply: Our study was designed to only measure cued-induced freezing by using a context during fear extinction and extinction recall that was new to the animals and very different from the conditioning context. The animals were also tested in another room. The high freezing at P24 probably reflects an age-related response to novelty. When extinction training started (i.e., during the first extinction block) animals of all ages and genotypes showed an increase in freezing relative to baseline (F(1,318)= 692.251, p < 0.001), including the P24 group (F(1,98)= 90.224, p < 0.001). This suggests that also in the P24 group freezing behavior reflected the recall of the learned the tone-shock association. It is clear that we were measuring freezing to the tone, since, again, the context used was very different from the one used during conditioning.
5. The authors should show all correlational analyses in a figure.
Reply: Please see Figure S3 (panels A-D) for the correlations between the molecular data and freezing behaviour.
6. The authors should specify the sex of the animals that were tested.
Reply: Only male rats were used, see ‘animals’ in materials and methods section.
The authors make several claims about the effects of age on extinction. To visualize these effects, the authors should graph the behavior of wild-type rats at each age, so direct comparisons can be made. An ANOVA on that data set should be done to confirm an effect of age.
Reply: We like to stress that the aim of the study was to investigate the effect of 5-HTT-/-on fear extinction across ages. However, to facilitate age comparisons, we have now included an additional figure in the supplement (Figure S1) depicting fear extinction learning across age in all genotype groups separately. We had already conducted ANOVA’s testing for the effects of age on extinction learning across blocks: “Exploration of the age effect revealed significant differences in extinction learning curves between all three ages, which seemed to be driven by slower extinction in pre-adolescent compared to adolescent rats (p = 0.006) and lower initial freezing (in block 1) of adult rats compared to preadolescent rats (p = 0.008). There were no age effects observed within the separate genotype groups (p > 0.1; Figure S1).”
On lines 472-473, the authors state that “the number of GAD65/67 positive cells…was decreased in the IL of 5-HTT-/- rats, regardless of age…” It is clear from Figure 2, that there is no significant difference between 5-HT+/+ and 5-HTT-/- rats in IL GAD at p70. The main effect of genotype is being driven by data from rats in the p24 and p35 groups. Similar mistakes are made throughout the manuscript.
Reply: An ANVOA revealed a main effect of genotype on the number of GAD65/67 cells, without a main effect of age and a significant age x genotype interaction effect. This is depicted in Figure 2. Notably, because there is no age x genotype effect (indicating that the effects of genotype are not age-dependent), it is statistically not allowed to conduct post hoctests at each individual age.
9. The inter-trial interval (ITI) during extinction testing was exceptionally short (5 seconds). A longer ITI is typical in rat studies and would have improved within-session extinction.
Reply: Protocols differ a lot in literature and there is no single protocol that serves as a ‘golden standard’. Instead, researchers develop their own protocols which work in their own hands, which is what we did. The protocol we used here (mass extinction with 24 CS presentations of 20 seconds and 5 seconds intervals), we also used on our previous fear conditioning and extinction papers with 5-HTT-/-rats (Shan et al., 2013, 2018, Schipper et al., 2018), which also report fear extinction (recall) impairments in 5-HTT-/-rats. Furthermore, our findings in rats correspond to fear extinction findings reported in 5-HTT-/- mice [30-36].
The authors should have assessed freezing during acquisition, as this can affect later levels of fear expression and extinction. Although the authors refer to previous work indicating that genotype does not affect learning, it is unclear whether that previous study tested the same three age groups.
Reply: We unfortunately did not assess freezing during acquisition (we did not record the video’s). We did in our previous work using the 5-HTT-/-rats, using the same fear conditioning protocol (Shan et al., 2018). In this study based on adult rats we found no genotype differences in freezing during acquisition. This, however, does not mean that freezing during conditioning is not different in preadolescent and adolescent rats. Since we cannot redo the whole experiment, we now mention this as a limitation of the study. Please note that freezing during blocks 1-4 during the fear memory recall/extinction session was not different between genotypes (Figure 1) and ages (Figure S1), which might suggest that initial fear acquisition is not affected by genotype or age.
11. It is difficult to follow the effects of the molecular analyses. The results of all post-hoc tests (multiple comparisons) should be clearly indicated with asterisks in the graphs.
Reply: We had already indicated the results of post hoctests for the effects of genotype at specific ages with asterisks in the graphs to make these findings more intuitive. All other effects are indicated with other symbols.
Round 2
Reviewer 2 Report
The authors now show data in the new supplementary figures that greatly improve the manuscript. However, the authors have not adequately addressed several of my concerns: 1. The authors continue to emphasize an effect on fear extinction recall when their most convincing effects are on extinction learning. As they point out, the enhanced freezing show by 5-HTT-/- rats on the second and third days of extinction MIGHT reflect impaired retention of extinction. However, it might also reflect a sustained impairment in extinction learning. Again, this impairment in extinction learning is what they do clearly demonstrate and is actually a strength of the manuscript. The potential effect on the recall of extinction should be discussed in the Discussion section, but is not something that is clearly demonstrated in this paper and therefore should not be emphasized to the extent that it is. The title should also be changed, since the impairment that is clearly demonstrated is on extinction learning, not extinction recall. 2. The authors argue that freezing that occurred during the first 2 minutes of the fear extinction context (pre-CS) reflects response to novelty rather than generalization of fear to a different context. As a result, they argue that this response would be the same as testing mice in the open field or elevated plus maze. I completely disagree. Open field testing and elevated plus maze testing are longer than 2 minutes in enclosures that are very different in size and shape than testing contexts in fear conditioning experiments. Importantly, open field and elevated plus maze testing rarely (if ever) evoke freezing responses. It is common for rats and mice to generalize their fear responses to another context even though the context is completely novel. If the authors want to argue that high freezing reflects an age-related response to novelty, then they should demonstrate this by testing response to novelty in non-fear conditioned animals of each age group. For example, do the authors see this age-related response when rats were exposed to the training context for the first time, before ever receiving a shock? 3. The authors have failed to provide graphs that allow the reader to visualize effects of age on extinction in wild-type rats at each age. 4. The authors maintain that the following sentence is accurate: “the number of GAD65/67 positive cells…was decreased in the IL of 5-HTT-/- rats, regardless of age…” As I stated in my previous review, the main effect of genotype is being driven by data from rats in the p24 and p35 groups, and not the p70 group. If each age group was analyzed separately, the conclusion of the same data would reveal that this decrease does not occur at each group. 5. It is unfortunate that freezing was not assessed during acquisition. However, mention of this as a limitation of the study improves the quality of the manuscript.Author Response
Dear editor,
Thank you for handling our manuscript entitled ”Impaired fear extinction recall in serotonin transporter knockout rats is transiently alleviated during adolescence”. We appreciate the thorough work of the reviewers. Reviewer 2 has some remaining comments. Please find our detailed responses to the comments below. New edits in the manuscript are indicated in green. We hope that our manuscript is now suitable for publication in Brain Sciences.
Rev. 2
1. The authors continue to emphasize an effect on fear extinction recall when their most convincing effects are on extinction learning. As they point out, the enhanced freezing show by 5-HTT-/- rats on the second and third days of extinction MIGHT reflect impaired retention of extinction.
However, it might also reflect a sustained impairment in extinction learning. Again, this impairment in extinction learning is what they do clearly demonstrate and is actually a strength of the manuscript. The potential effect on the recall of extinction should be discussed in the Discussion section, but is not something that is clearly demonstrated in this paper and therefore should not be emphasized to the extent that it is. The title should also be changed, since the impairment that is clearly demonstrated is on extinction learning, not extinction recall.
Reply: We understand the point of the reviewer, definitely considering the shape of the graphs depicted in figure 1 of the manuscript. However, the claim of genotype x age interaction effects on extinction learning are not supported by the statistics, as we have set out in the first rebuttal:
“We agree with the reviewer that the on average increased freezing levels during the first fear extinction recall session could potentially be explained by impaired initial fear extinction learning. However, if this would be the case, one might expect the 5-HTT-/-rats to catch up in fear extinction during the presentation of subsequent cues, as tested for in the second and third fear extinction session. However, in these two sessions, no significant block x genotype effects were observed (recall session 1: p = 0.123; recall session 2; p = 0.549), nor block x age x genotype interactions (recall session 1: p = 0.272; recall session 2; p = 0.847), indicating that genotype did not modulate extinction learning across blocks in these sessions. However, there were significant main effects of genotype (recall session 1: p = 0.018; recall session 2: p = 0.001), as well as genotype x age effects (recall session 1: p = 0.032; recall session 2: p = 0.039), suggesting that genotype affected freezing behaviour across the whole session, independent of block. Therefore, we believe that the increased freezing behavior as observed in pre-adolescent and adult 5-HTT-/-rats is not the mere result of impaired extinction learning, but instead indicates additional effects of impaired extinction recall. To support this point, we included an additional figure (Figure S2) to the revised manuscript depicting the freezing across blocks in the recall sessions”.
Even the data from the first extinction (learning) session indicates that the genotype x age interaction effect, cannot be explained by altered extinction learning across blocks (genotype x age x block: F(20, 1308)< 1). However, we cannot exclude differences in extinction learning either. We therefore now mention it in the discussion section of the manuscript.
2. The authors argue that freezing that occurred during the first 2 minutes of the fear extinction context (preCS) reflects response to novelty rather than generalization of fear to a different context. As a result, they argue that this response would be the same as testing mice in the open field or elevated plus maze. I completely disagree. Open field testing and elevated plus maze testing are longer than 2 minutes in enclosures that are very different in size and shape than testing contexts in fear conditioning experiments. Importantly, open field and elevated plus maze testing rarely (if ever) evoke freezing responses. It is common for rats and mice to generalize their fear responses to another context even though the context is completely novel. If the authors want to argue that high freezing reflects a related response to novelty, then they should demonstrate this by testing response to novelty in non-fear conditioned animals of each age group. For example, do the authors see this agerelated response when rats were exposed to the training context for the first time, before ever receiving a shock?
Reply: Thank you for this discussion point. We did not mean to say that with the 2 min habituation period we replicate exactly the open field test or the elevated plus maze test, but the response to novelty. During the 2-min habituation period prior to CS+ exposure in the fear extinction test (session 1) the animals are placed in a novel environment. Hence, it is plausible to consider the freezing during this 2-min habituation period as a response to novelty. Please note that we did not use the comparison with the open field test and elevated plus maze test in the manuscript (we mentioned it in the rebuttal only). In the manuscript we describe it as baseline freezing in a novel context, exactly what it is without further interpretations.
3. The authors have failed to provide graphs that allow the reader to visualize effects of age on extinction in wild-type rats at each age.
Reply: We have provided the graph, it is available as figure S1, in the supplementary document.
4. The authors maintain that the following sentence is accurate: “the number of GAD65/67 positive cells…was decreased in the IL of 5-HTT-/- rats, regardless of age…” As I stated in my previous review, the main effect of genotype is being driven by data from rats in the p24 and p35 groups, and not the p70 group. If each age group was analyzed separately, the conclusion of the same data would reveal that this decrease does not occur at each group.
Reply: We indeed did not change our statement because of the statistics. We conducted a two-way ANOVA, revealing a genotype effect but no age x genotype effect, indicating that the effect of genotype was not significantly different between the distinct age groups. Therefore, no further testing is warranted. However, as we agree that the effect of genotype is primarily prominent at the preadolescent and adolescent age groups, we now included the post hoctests the reviewer is requesting. The section now reads:
“The number of GAD65/67 immunopositive granules in the IL was significantly affected by genotype (F(1, 24)= 14.326, p = 0.001), but not age (F(2, 24)= 2.110, p = 0.143), and no genotype x age interaction could be detected (F(2, 24)= 1.222, p = 0.312, (Figure 2). The number of granules expressing GAD65/67 was significantly reduced in 5-HTT-/-animals compared to 5-HTT+/+animals (p = 0.001). Although the effect of genotype did not significantly differ between age groups, post hoctesting revealed most prominent effects of genotype in preadolescent rats (p < 0.001), whereas adolescent and adult rats did not display significant differences between genotypes (p’s > 0.3).”
5. It is unfortunate that freezing was not assessed during acquisition. However, mention of this as a limitation of the study improves the quality of the manuscript.
Reply: Thank you for this comment. It is mentioned as limitation in the discussion. It is indicated in bold red from the previous revision